# Exploring the Influence of Perceived Ingroup and Outgroup Threat on Quality of Life in a Region Impacted by Protracted Conflict

**DOI:** 10.3390/ijerph20166599

**Published:** 2023-08-18

**Authors:** Izzeldin Abuelaish, Ayesha Siddiqua, Susan J. Yousufzai, Caroline Barakat

**Affiliations:** 1Global Health Division, Dalla Lana School of Public Health, University of Toronto, Toronto, ON M5T 3M7, Canada; susan.yousufzai@utoronto.ca; 2Department of Health Research Methods, Evidence and Impact, McMaster University, Hamilton, ON L8S 4L8, Canada; siddia36@mcmaster.ca; 3Faculty of Health Sciences, Ontario Tech University, Oshawa, ON L1G 0C5, Canada; caroline.barakat@ontariotechu.ca

**Keywords:** ingroup threat, intragroup conflict, intergroup conflict, outgroup threat, psychological wellbeing, political conflict exposure, perceived threat, quality of life

## Abstract

While the detrimental effects of protracted political conflict on the wellbeing of Palestinians living in the occupied Palestinian territory (oPt) are generally recognized, the impact of perceived threat on quality of life (QoL) faced from within their community (ingroup; Palestinians) and from the outgroup (Israelis) is unexplored. This cross-sectional study examined the following: (1) The status of perceptions of QoL on four domains measured by the World Health Organization Quality of Life (WHOQoL-Bref) instrument, physical health, psychological health, social relationships, and environment, among Palestinian adults (*n* = 709) living in the Gaza Strip; (2) The associations between perceived ingroup threat (PIT) and QoL on the four domains; (3) The associations between perceived outgroup threat (POT) and QoL on the four domains. Multivariable linear regression models revealed PIT was negatively associated with QoL in each of the four domains (*p* < 0.001). POT was positively associated with QoL in three of the four domains: physical health (*p* < 0.001), psychological health (*p* < 0.001), and social relationships (*p* < 0.001). This study contributes valuable insights into how QoL is viewed by a group experiencing collective existential threat. The findings expand the limited recognition of the reciprocal roles of perceived threat from the ingroup and outgroup on the QoL of vulnerable populations.

## 1. Introduction

The Israeli–Palestinian conflict resulted in the Israeli capture and occupation of several Arab territories, including the West Bank and the small yet densely populated Gaza Strip—now recognized by the United Nations as occupied Palestinian territories (oPt) [1,2]. For close to a century, Palestinians residing in these occupied territories have been experiencing trauma due to war and ongoing political violence [1,3]. This war is a prime example of an intractable (ethnonational) conflict characterized by extreme intergroup violence, which has constituted a very real source of collective, existential threat to the Palestinian population [4]. The threat Palestinians face from the Israelis—considered as the outgroup—is a key negative intergroup stressor which has led to adherence to the ethos of conflict for both of these groups [5,6]. The entrenched culture of conflict has undergone extensive institutionalization, manifesting itself in a relentless cycle of feedback, whereby the perception of threat arising from exposure to political conflict fuels the incitement of fear, violence, and warfare [7,8,9]. In turn, this begets a spiraling dynamic of hatred, which engenders further fear, violence, and escalation of the conflict.

Empirical evidence on threat perceptions in intergroup conflict has primarily focused on the implications of these conflicts on the types of perceived threat (realistic vs. symbolic) for intergroup relations, showing outgroup threat to play a central role in driving parochial altruism and ingroup cohesion [10,11,12,13], while fostering prejudice and bolstering existing schemas about the enemy [14,15,16]. In the context of conflict, perceived threat often leads to the magnification of existing cognitive biases that arise naturally out of mere group categorization with one’s ingroup [17,18,19,20]. This phenomenon is shaped by inherent similarities influenced by shared beliefs that serve as guiding principles for our actions [19,20]. Such perceptions stem from the benefits and fulfillments derived from group membership, such as acceptance, belonging, and social support [18,20,21,22]. Since they hold significant value for us, we fear our group’s dissolution almost as much as we fear our own demise; thus, we tend to attribute favoritism towards our own group [18,22]. As individuals become more entrenched in their opinions and less receptive to the perspectives of others who have opposing beliefs, this can result in heightened polarization and increased hostility between groups. 

When examining the concept of perceived threat within group dynamics, it is essential to distinguish between the factors associated with perceived threat from the ingroup and the outgroup in conflict situations—an aspect which has not received adequate attention in previous research conducted in this field. Perceived intergroup threat is conceptualized as the belief that a given out-group’s actions are in some way inimical to one’s in-group [14,16,17]. Integrated threat theory (ITT) distinguishes two types of perceived threats. Threats to a group’s political influence, economic dominance, or bodily security constitute realistic threats [17]. Whereas, threats to a group’s doctrine, philosophy, morality, or worldview are considered symbolic threats [17,18] and have been considered a stronger predictor of intergroup hate [9]. In the context of the Israeli–Palestinian conflict, the conceptualization of outgroup threat as perceived by the Palestinians is posed by the Israelis due to past, present, and anticipation of future harmful relations [13,18,23]. The perceived threat constitutes tangible concerns related to physical safety, material wellbeing, competition for power, resources, and territory [6,18]. It also encompasses symbolic threats towards the Palestinian’s values and beliefs, and a collective threat to their national identity [18,23]. 

The range of threats experienced by Palestinians residing in the oPt also stems from their ingroup (based on a shared national identity)—where members perceive threat internally from within their own community. At the ingroup level, differences in political or religious ideologies can emerge and induce conflicts between opposing ideological subgroups within the same ingroup [15]. This phenomenon of affective political polarization and heightened animosity between partisan groups, sometimes even described as an intractable internal conflict, deepens the societal division among ideological subgroups [15]. Although the Palestinians have a shared interest in a patriotism centered around the struggle against their adversary, driven by a love for their homeland, the pursuit of national Palestinian goals has generated political discourse between different factions—ensuing an intra-Palestinian conflict [23,24,25]. The extreme political polarization in the oPt has resulted in domination of areas in the West Bank and Gaza Strip by large Palestinian factions, including the national secular Fatah and religious Islamic movement Hamas [6]. The Palestinian Authority (Fatah), which controls the West Bank, acknowledges the State of Israel and is perceived by Israelis as willing to engage in a peace process based on a two-state solution, whereas the Hamas government is opposed to the peace process [6]. Due to the ideological, sociological, and geographical divisions between the Gaza Strip and the West Bank, resulting from the Palestinian civil war in 2007 between Hamas and Fatah, it is plausible to assume that perceived threat from the ingroup may be apparent due to uncertainty around actions of the different factions [6,26]. 

Hatred is found to develop as a result of abstract and long-term threats that are centered around personal values and identity, signifying that symbolic threats according to the ITT, may be particularly sensitive to hatred [9,26]. The negative emotions associated with hatred cultivate hatred, fear, and threat, which often have negative downstream consequences and increase the risk of adverse health outcomes [7,26,27]. The perception of threat from an outgroup has been connected to both the experience of physical stress responses (such as salivary cortisol levels) [27] and negative opinions about the outgroup [28]. In relation to conflict, numerous studies demonstrate that long-term exposure to political violence can cause severe psychological distress, such as constant emotional and physiological arousal, which includes heightened anxiety, feelings of reduced safety, symptoms of posttraumatic stress, and a subjective sense of insecurity [14,29,30,31,32]. However, limited research has reported on the health and wellbeing of Palestinians in the Gaza Strip, as a consequence of perceived collective, existential threat to their national identity [13,23,31,33]. Moreover, the majority of the extant literature focuses on psychological health outcomes associated with political conflict exposure [4,14,34,35,36,37]. For example, Sousa et al. [4] found that home invasions and demolition led by Israelis had significant implications for Palestinian women’s emotional wellbeing. In a study conducted by McNeely et al. [36], utilizing a representative sample of Palestinians in the oPt, it was discovered that human insecurity, characterized by feelings of fear regarding the safety of one’s home, family, and oneself, as well as resource inadequacy (e.g., insufficient housing, clothing, household amenities, food, recreation, and transportation), were associated with experiences of depression and trauma-related stress.

To comprehensively assess the various dimensions that define quality of life, it is necessary to conduct research that takes into account diverse contextual factors. The World Health Organization describes quality of life (QoL) as, “individuals’ perceptions of their position in life, that is rooted in the context of the culture and value systems in which they live, and in relation to their goals, expectations, standards and concerns” [37]. This involves examining factors associated with physical and psychological health, social relationships, and the environment where QoL is derived. Studies conducted over a decade ago in the oPt measuring QoL as determined by the World Health Organization quality of life (WHOQoL-Bref) instrument found that the physical, psychological, and environment domain scores were among the lowest in comparison to 23 other countries included in the WHO International Field Trial [25,37]. In the oPt, lower QoL was linked with men, older individuals, and those with lower levels of education compared to their counterparts [25]. In addition, QoL scores were negatively associated with higher levels of distress and fear, as well as lower financial status and freedom [25]. Conversely, while outgroup threat may have negative psychological impacts on vulnerable populations and individuals, some research shows that upon perceiving a threat from the outgroup to their ingroup, individuals tended to exhibit an increased strength of identification with their ingroup [11,12,14]. This heightened identification may indirectly contribute to a positive effect on QoL by strengthening social relationships within the ingroup. 

Nonetheless, there is a paucity in literature exploring the broader health implications associated with the political conflict in the oPt. To our knowledge, no study has examined a nationally representative sample, while considering the relationship between perceived threat from the ingroup and outgroup on health-related outcomes. This is crucial as threat perceptions play a key role in violent intergroup conflict settings [14,15,29], and perceived threat in this ongoing political conflict setting may not be eminent from the outgroup exclusively. The differential perception of perceived ingroup versus outgroup threat may have distinct consequences and effects on the wellbeing of individuals living in the oPt. 

The objectives of this study were to examine the following: The status of perceptions of QoL on the four domains: physical health, psychological health, social relationships, and environment among Palestinian adults living in the Gaza Strip (for the overall cohort and by demographic characteristics);The associations between perceptions of threat from the ingroup and perceptions of QoL on the four domains: physical health, psychological health, social relationships, and environment;The associations between perceptions of threat the from outgroup and perceptions of QoL on the four domains: physical health, psychological health, social relationships, and environment.

## 2. Materials and Methods

### 2.1. Study Design and Setting 

This was a cross-sectional study using data collected from the community-based survey “Developing a Measure of Hatred and its Impact on Health and Wellbeing”, which was implemented from May 2019 until June 2019 among Palestinian adults living in the Gaza Strip. Ethics approval from the University of Toronto Research Ethics Board (protocol ID: 37953) was obtained. 

### 2.2. Sample Size and Sampling 

EpiInfo [38] was used to calculate sample size for this study. Simple random sampling was used for this cross-sectional study. The following parameters were used for the sample size calculation: Z = 2.57 (corresponding to a 99% confidence interval) and c = 5% (corresponding to the margin of error) in the formula n = [(Z^2^) × p × (1 − p)]/ (c^2^), where n = sample size. Since there were no previous estimates of the prevalence of hatred or wellbeing in this population, there were no two contrasting groups to compare. Thus, maximum variation was assumed for the estimate of hatred and wellbeing in the population to calculate the largest possible sample size, given the aforementioned parameters. This yielded a required sample size of about 660. Assuming that there may be a maximum of 25% refusal rate, 825 participants were required in the study. 

### 2.3. Study Population 

Palestinian adult males and females living in all five gouvernantes (North, South, Middle, Khan Younis, and Gaza) of the Gaza Strip aged 18 to 60 years were considered for inclusion. Those under 18 years of age and not able to read or write or both were excluded from the study. Local Community Health workers volunteered to assist in data collection and were trained by the principal investigator in community mapping and enumerating households in the Gaza Strip. Volunteers visited each household and designed a schematic map symbolizing each household and identifiable landmarks. During the process of mapping, households were visited to note the presence of any eligible members. The eligible household and number of eligible participants were used to create a sampling frame. A random number table was used to draw the eligible households from the sampling frame. Eligible households were administered the survey and were read out the consent forms, ensuring all participants had a solid understanding of the study. All participants were required to sign a consent form to participate.

### 2.4. Study Measures 

The survey used in this study was a validated double translated collection of measures aimed at identifying the impact of exposure to protracted violence, blockade, and multiple military attacks. All measures that were included in the test battery were validated by judges through face validity and translated (and back translated) from English to Arabic. The entire test battery consisted of 334 items and took approximately 45 min to complete. 

### 2.5. Exposures

#### 2.5.1. Ingroup Identification/Outgroup Hatred

The exposures were perceptions of threat from the ingroup (Palestinians) and outgroup (Israelis), which were measured using a scale created based on findings of a study examining ingroup identification/outgroup hatred in the United States by Cottrell and Neuberg [39], adapted from Mackie et al. [40]. This scale assessed threats a person perceives towards their own group or way of life from others. This includes threats to job and economic opportunities, personal possessions, rights and freedoms, physical health, and safety, ingroup values, and morality. These items were adapted to reflect the Palestinian-Israeli conflict (Appendix A). This scale captured differences in threat perception in terms of how a person rated the threat posed by a member of their ingroup (Palestinians) versus a member of their outgroup (Israelis), where threat from each group is assessed using a set of 9 questions, respectively, using a 5-point Likert scale. Examples of these questions are “Palestinians threaten my people’s personal freedoms and rights” and “Israelis are a threat to my people’s values”. For questions that pertain to threat posed by a member of the participant’s ingroup, a higher score indicates higher perception of threat from the ingroup. For questions that pertain to threat posed by a member of the participant’s outgroup, a higher score indicates higher perception of threat from the outgroup. 

#### 2.5.2. Outcomes 

The outcomes were perceptions of quality of life on four domains: physical health, psychological health, social relationships, and environment, which were measured using the Abbreviated World Health Organization Quality of Life (WHOQoL-Bref) instrument. The WHOQoL-Bref is a cross-culturally validated measure with 26 items using a 5-point Likert scale [37]. Diverse facets of an individual’s life are assessed in each of the four domains. The physical health domain includes items regarding activities of daily living and work capacity, the psychological health domain includes items regarding negative and positive feelings, the social relationships domain includes items regarding personal relationships and social support, and the environment domain includes items regarding home and physical environments—among many other related items in each of these domains. The four domain scores indicate an individual’s perception of QoL in each particular domain. Domain scores are scaled in a positive direction, where higher scores indicate higher QoL [37]. The Cronbach’s alpha for the domains ranged from 0.68 to 0.85 in previous samples; this assessment has been deemed a valid and reliable measurement. 

#### 2.5.3. Covariates 

From a social epidemiology and public health perspective, each of the demographic characteristics included in this study were hypothesized to be potentially related to perceptions of QoL on each of the four domains measured by the WHOQoL-Bref. Thus, the impact of perceptions of threat from the ingroup (Palestinians) and outgroup (Israelis) were examined as exposures while controlling for all demographic characteristics as plausible covariates.

### 2.6. Analysis 

Data were analyzed using SPSS version 28. Descriptive statistics, including frequencies and percentages, summarized the demographic characteristics. To address the study objective 1, the mean (SD) summarized the status of perceptions of quality of life on each of the four domains measured by the WHOQoL-Bref for the overall cohort and by demographic characteristics. To address study objectives 2 and 3, multivariable linear regression models were used to determine the associations between the perception of threat from the ingroup and perceptions of QoL on the four domains, as well as associations between the perception of threat from the outgroup and perceptions of QoL on the four domains. For each of the eight modelled outcomes, all independent variables were entered into the model simultaneously. For each categorical covariate variable, one category was chosen to be the reference category, with each category of the variable then compared to the reference. This process resulted in models that determined the significant associations between perceptions of threat from the ingroup and outgroup and perceptions of QoL on the four domains, after controlling for all demographic characteristics entered into the models. Statistically significant associations were tested at *p* < 0.05. An adjusted R square was used to determine the goodness-of-fit for each of the models. An F-test was conducted to determine the statistical significance of the models in predicting the outcomes. The key assumptions of multivariable linear regression were met in our models, including homoscedasticity, normality of residuals, and no multicollinearity between the independent variables. For the multivariable linear regression models, at least 20 observations were required for each independent variable included in the model [41]. 

## 3. Results

The study employed a large random sample of adult Palestinians (*n* = 1200, 59% response rate). There were 709 participants in this study—their demographic characteristics are summarized in Table 1. Overall, 333 (47.1%) participants in this study were male. The majority of the participants were Muslim. A total of 347 (50.4%) participants were married, and 358 (50.5%) participants had children. A total of 178 (27.1%) participants were members of political parties aligned with their personal values, with majority membership to the political party Fateh (15.2%). Gouvernantes were combined based on the following regions: North (North and Jabalia), Middle (Dier al-Balah), and South (Rafah and Khan Younis). A total of 333 (47%) participants lived in the North Area, whereas 254 (35.8%) and 121 (17.1%) participants lived in the South and Middle Areas, respectively. A total of 339 (47.8%) participants had a university bachelor’s degree. A total of 405 (67.2%) participants were unemployed. A total of 366 (55.5%) participants reported the perception of not living in a peaceful area.

### 3.1. Perceptions of Quality of Life

Table 2 shows perception of QoL on the four health domains (physical health, psychological health, social relationships, and environment) according to demographic characteristics. The mean score for perception of QoL in the physical health domain was 13.40 (SD: 2.76). The mean score for perception of QoL in the psychological health domain was 12.90 (SD: 2.93). The mean score for perception of QoL in the social relationships domain was 13.82 (SD: 3.68). The mean score for perception of QoL in the environment domain was 11.71 (SD: 2.85). 

### 3.2. Associations between Perception of Threat from Ingroup and Perceptions of Quality of Life 

As the score for the perception of threat from the ingroup increased, the score for the perception of quality of life decreased in each of the four domains measured using the WHOQOL-Bref: physical health (B: −0.06; 95% CI: −0.08, −0.04; *p* < 0.001), psychological health (B: −0.07; 95% CI: −0.09, −0.05; *p* < 0.001), social relationships (B: −0.06; 95% CI: −0.09, −0.03; *p* < 0.001), and environment (B: −0.04; 95% CI: −0.06, −0.02; *p* < 0.001) (Table 3). 

### 3.3. Associations between Perception of Threat from Outgroup and Perceptions of Quality of Life 

As the score for the perception of threat from the outgroup increased, the score for the perception of quality of life increased in three of the four domains measured by the WHOQOL-Bref: physical health (B: 0.05; 95% CI: 0.03, 0.07; *p* < 0.001), psychological health (B: 0.05; 95% CI: 0.02, 0.07; *p* < 0.001), and social relationships (B: 0.07; 95% CI: 0.04, 0.10; *p* < 0.001) (Table 4). 

## 4. Discussion

This paper sought to examine the status of perceptions of quality of life on four domains, physical health, psychological health, social relationships, and environment among Palestinian adults living in the Gaza Strip, as well as the relationships between perceived threat from the ingroup and outgroup within these domains of quality of life. In contrast to the findings reported in the WHOQoL-Bref international trial, which examined mean domain scores from 23 countries [42], the participants in this study displayed significantly lower scores across all domains. Specifically, the scores were lower in the domains of physical health (13.4 vs. 16.2), psychological health (12.9 vs. 15.0), social relationships (13.8 vs. 14.3), and environment (11.7 vs. 13.5). When compared to health domain scores obtained from a general population in Iran, the scores in the domains of physical health (14.3), psychological health (13.4), and environment (12.3) were still higher than in the current study [43]. However, the social relationships domain showed a similar score (13.9) [43]. Consistent with a previous study conducted in 2009 among Palestinians in the oPt [25], the social domain scores remained relatively high compared to the other health domains in the current study. These findings underscore the distinct and notable disparities in the participants’ QoL in relation to the broader international and regional contexts.

Perception of threat from the ingroup was associated with a lower QoL on all health domains, while perception of threat from the outgroup had a significant positive association on the domains pertaining to physical health, psychological health, and social relationships. Our research suggests that perceived threat from the ingroup is likely to have negative impacts on health, considering physical health, psychological health, social relationships, and environmental health indicators, and extends prior research which has primarily focused on consequences of perceived intergroup threat on psychological wellbeing [14,29,30,33,34,35].

While numerous Palestinians have been forced to leave their residences and ancestral land, there are still many who remain steadfast and deal with constant threat to the security, standard, and safety of their living conditions [4]. Embedded in greater Palestinian society, both locally and in the diaspora, nationalism, and confrontation with external political powers can reinforce reliance and interdependence between the individual and the community, which serves as a buffer in times of stress [44]. Thus, ingroup threat is particularly detrimental to the health of this population, as they are heavily dependent on the solidarity of their ingroup as a means of surviving the negative surrounding context. This is one reason we may see a negative relationship between perceived ingroup threat and QoL on all domains in this study. Social identity theory suggests that people associate themselves with social groups, which helps them build self-esteem and gain access to social resources to deal with life’s difficulties [45]. By being part of a social group that one closely identifies with, individuals not only mobilize support networks but also develop effective ways of coping with adversity [45,46]. In this way, perceived threat from the ingroup can adversely impact QoL, eroding perceived ingroup stability and legitimacy, and fading any psychological benefits [45]. 

Tropp and Wright [47] showed that the ingroup becomes included in the self with higher ingroup identification. Thus, perceived threat from the ingroup may seem like a threat to one’s self identity, negatively impacting QoL. This result is in accordance with the antecedents of threat in ITT, suggesting that stronger identification with the ingroup—particularly when the identity is salient and important [17,18,47,48]—can sensitize individuals to the negative effects of ingroup threat [46]. This phenomenon has been linked to the idea of symbolic individual threats, that involve the risk of losing reputation or honor, as well as the erosion of an individual’s sense of identity or self-worth [15,17,22]. For example, people may experience greater distress when a group to which they belong (ingroup) is perceived to have committed an immoral act and may lead to a reduction in identity affirmation. Indeed, one study found that individuals may exhibit hostility towards an ingroup that violates social norms, triggering feelings of shame and a negative self-image [49]. 

It is well known that stress can increase vulnerability to illness by lowering immune system functioning. Thus, in a state of continuous exposure to threat and fear, heightened negative emotions, and increased levels of cortisol due to expression of the body’s stress response may adversely impact psychological and physical health [14,27]. For example, evidence suggests negative emotions such as depression, anxiety, anger, and hostility tend to contribute to mortality and morbidity, by disrupting the immune system and increasing the risk of various non-communicable diseases (e.g., cardiovascular disease, arthritis, certain cancers, and Alzheimer’s disease) [50,51]. Indeed, studies exploring the physiological consequences of intergroup threat have linked discrimination from outgroups among racialized minorities to the pathogenic sequelae of stress reactivity such as high blood pressure, changes in cardiovascular patterns, and an increase in cortisol concentration [27,40,51,52]. Although these studies have looked at intergroup relations, further research is warranted on physiological outcomes, in relation to intragroup perceived threat, which may have greater consequences in certain cultural contexts, as seen in this study. 

Our results align with realistic group conflict theories suggesting that conflict with outgroups fosters group cohesion, where group cohesion may increase when groups are threatened [53]. Thus, the positive association between perceived threat from the outgroup and the social relationships domain may be explained by the greater ingroup cohesion that emerges as a result of external threats [10,11,12]. One study found that while perceived threat can have direct negative consequences for psychological health it can also lead to greater social identification and, consequently, exert positive indirect effects on wellbeing [14]. Threat can also induce positive behaviors, which are sometimes motivated by wanting to appear non-prejudiced (i.e., maintain positive image of self and the ingroup) [16]. Indeed, the ability to cope with injustice and discrimination and attenuate the effect of political conflict exposure on psychological distress is predicted by social identification [45] and ideological belief [28,34]. This behavior has been attributed to the concept of embracing the ethos of conflict [28], which implies that societies, in order to effectively deal with the negative repercussions of ongoing perceived threats, adopt beliefs that promote and justify conflict. These beliefs serve to provide them with a meaningful framework for comprehending the conflict, thereby alleviating their sense of uncertainty and stress [28,54]. Palestinians are constantly affected by discrimination and injustice, so those who have higher identification with their group may cope better, having a positive indirect impact on QoL.

Perceived outgroup threat can increase social identification and make the Palestinian identity more salient which would increase norms and health-related behaviors related to *Sumud*—a national concept and psychological strategy which has preserved them to their land in fight for national liberty; it is regarded as steadfastness and the determination not to leave or be violently removed from their land [44]. In the context of collective, existential threat, a positive association between perceived outgroup threat and physical health, psychological health, and social relationships may be influenced by the resilience within the broader ecological, social, and cultural context of the Palestinian population [32,44]. Social and cultural context, family unity, solidarity, and a sense of coherence, which are often less idolized in Western understanding, play crucial roles in comprehending the resilience of people in these conflicted areas and buffering against distress-related pathology [55]. 

During sensitive periods of development, such processes can alleviate the negative health impacts of living in a protracted conflict region. For instance, children in deprived neighborhoods are able to thrive despite the vulnerabilities they face, due to the sense of cohesion they feel in their neighborhood [56]. For example, Veronese et al. [3] suggest that Palestinian children resort to spatial agencies, such as their homes and schools, to embody resistance to preserve positive function and subjective wellbeing in the face of disaster. Evidence also suggests that individuals with low socioeconomic status confront the adversities they face by using ‘shift-and-persist’ strategies to alleviate stress and adapt the self through reappraisals, while holding on to meaning and optimism, which helps to attenuate sympathetic-nervous-system and hypothalamic–pituitary–adrenocortical (HPA) responses [57]. 

In the current political climate, it is crucial to find ways to foster resilience, given limited resources available to enhance the material and economic resources of people in the oPt [1,55]. Indeed, research shows that the wellbeing of individuals in the oPt is significantly impacted by realistic threats imposed on their livelihood, which has resulted in a loss of resources, a lack of security, and chronic economic constraints leading to poverty and unemployment [25,58,59]. Despite their high educational status, half of the participants in this study reported unemployment. Given the significant poverty rates and the unstable political climate in the oPt, it is imperative for the international community to intervene and facilitate access to employment opportunities, mental health support, and social services, which are crucial for maintaining the resilience and wellbeing of the population [32,35]. The long-standing and entrenched conflict, spanning generations, has undeniably contributed to a decline in QoL. Furthermore, considering the prediction made by the United Nations [24] that Gaza would become uninhabitable by 2020—a prediction that has currently been realized due to rapidly deteriorating living conditions—it becomes vital to closely monitor the QoL of this vulnerable population [35].

It is also important to consider that, while outgroup threat can bolster saliency of the Palestinian identity to cope with the conflict [5], this may create a paradoxical situation in which the heightened strength in identity may adversely impact the psychological health of individuals who identify more strongly with their transgressing ingroup, as it may act as a negative reflection of themselves [49]. Moreover, although stress responses may serve as a protective mechanism in the short term, repeated exposure to stressors can the increase allostatic load that may pose a risk for negative long-term health outcomes [27]. It is crucial to acknowledge the threat that is pervasive within and outside the community of this population, as it is evident that long-term consequences and presence in this conflict may have devastating effects on their QoL, and the increasing uncertainty may lead to greater divide, discordance, and polarization. 

### Limitations and Future Directions

A key limitation of this research study is that the data are cross-sectional, preventing us from drawing confident conclusions on the nature of causality. However, the study benefits from a large general population sample, which is usually challenging to obtain, particularly when dealing with a sensitive topic like this. Even though there are limitations, the results of the study provide essential insights. In uncovering the interplay between perceived outgroup and ingroup threats, among a large general population sample in a setting that has experienced extreme intergroup conflict, our research highlights the importance of considering that perception of outgroup threat is not the only factor to contend with for people living in protracted conflict regions. 

This study recommends future research to employ longitudinal designs, which can examine the differences in consequences associated with perceived outgroup and ingroup threats as well as differentiate between conceptually distinct threats (e.g., symbolic vs. realistic). A lack of security, which is usually given by ingroup cohesion, leads to frustration, dissatisfaction, and possibly a self-reinforcing cycle of ingroup threat and lower social cohesion that increase threat and hatred [41,54]. Moreover, it is believed that when the entire ingroup is threatened a more probable adaptive reaction, such as anger, is theorized to emerge in response to fear. This anger response can potentially motivate the ingroup to address the threat [17,18]. However, this reaction is likely to perpetuate a harmful cycle of increased intolerance, intergroup hostility, and animosity [18,23]. Correspondingly, a causal relationship may exist in relation to the direct or indirect exposure of such triggers and hatred as an outcome, warranting further exploration [23,26]. For instance, given that hatred, is often a consequence of threat [9,18,60,61], and fear is a product of incitement from protracted conflict, and hatred and threat are known to induce the body’s stress response, future research may want to explore the link between perceived threat and hatred, and the spread of this within the community, as well as the associated health risks from both ingroup and outgroup hatred using objective and direct indicators of health (e.g., physiological, inflammatory, and neural markers). To address the impact of threat on hatred and their consequences on health in both intergroup and intragroup conflicts, it is necessary to use an interdisciplinary and multi-disciplinary approach by combining the biopsychosocial and public health perspectives, to holistically examine the socio-epidemiological and pathophysiological aspects of this phenomenon. It is also important to develop effective strategies for conciliation, management, and mitigation of adverse health outcomes in regions impacted by protracted conflict. 

## 5. Conclusions

The findings of this study underscore a necessary element of perceptions of intergroup threat and step towards understanding the reciprocal impact of perceived ingroup and outgroup threat, which may have distinct consequences and effects on the wellbeing of individuals living in regions of protracted conflict. The present key findings show that the perception of threat from the ingroup has a negative impact on all QoL health domains, while the perception of threat from the outgroup is positively associated with physical health, psychological health, and social relationships. This suggests that an increase in perceived QoL can serve as a natural adaptive response when a group faces a collective threat. However, it is important to consider the long-term implications in this political conflict context - where collective, existential threat persists - potentially leading to the eventual breakdown of these same adaptive defenses. The findings also emphasize the importance of understanding the differential impact associated with perceived threat on health in group phenomenon, and its potential relevance in non-conflict settings. 

## Figures and Tables

**Table 1 ijerph-20-06599-t001:** Demographic characteristics of study participants (n = 709).

Demographic Characteristics	
Gender n (%)	
Male	333 (47.1)
Female and missing *	376 (53)
Age (years) mean (SD)	29.8 (9.6)
Missing	18 (2.5)
Religion n (%)	
Muslim	687 (96.9)
Christian	7 (1.0)
Missing	15 (2.1)
Marital status n (%)	
Single	290 (40.9)
Married	347 (48.9)
Divorced	28 (3.9)
Widow	23 (3.2)
Missing	21 (3.0)
Has children n (%)	
Yes	358 (50.5)
No	289 (40.8)
Missing	62 (8.7)
Member of political party that aligns with personal values n (%)	
Yes	178 (25.1)
No	478 (67.4)
Missing	53 (7.5)
Political party membership n (%)	
Fateh	108 (15.2)
Hamas/Jehad	31 (4.4)
Left parties (Democratic Front, Popular Front, And Palestinian Front)	18 (2.5)
Others, independent, cannot tell, and missing *	552 (77.9)
Area of residence n (%)	
North Area (North Region and Gaza)	333 (47.0)
Middle Area (Dier al-Balah)	121 (17.1)
South Area (Rafah and Khan Younis) and missing *	255 (35.9)
Highest level of education completed n (%)	
Primary school	11 (1.6)
Middle school	28 (4)
High school	155 (21.9)
Diploma or college	100 (14.1)
University bachelor’s degree	339 (47.8)
Technical or community college	7 (1)
Masters	20 (2.8)
PhD and missing *	49 (6.9)
Employment status n (%)	
Unemployed	405 (57.1)
Retired	13 (1.8)
Full-time government job	60 (8.5)
Part-time government job	17 (2.4)
Full-time job (private sector)	55 (7.8)
Special and part-time job (private or personal)	53 (7.5)
Missing	106 (15.0)
Income currency n (%)	
Shekel	220 (31)
Dollar	37 (5.2)
Jordanian dinar and missing *	452 (63.8)
Perception of living in a peaceful area n (%)	
Yes	293 (41.3)
No	366 (51.6)
Missing	50 (7.1)

* Cell sizes < 6 suppressed; missing data for each variable is indicated as n (%).

**Table 2 ijerph-20-06599-t002:** Status of WHOQOL-BREF domain scores according to demographic characteristics.

WHO Domains	Physical Health Domain Scores Mean (SD)	Psychological Health Domain Scores Mean (SD)	Social Relationships Domain Scores Mean (SD)	Environment Domain Scores Mean (SD)
Overall/cohort	13.40 (2.76) (n = 693)	12.90 (2.93) (n = 692)	13.82 (3.68) (n = 692)	11.71 (2.85) (n = 694)
Demographic characteristics				
Sex				
Male	13.17 (2.72)	12.54 (3.01)	13.24 (3.89)	11.63 (2.87)
Female	13.61 (2.78)	13.22 (2.83)	14.37 (3.37)	11.79 (2.83)
Religion				
Muslim	13.40 (2.77)	12.91 (2.93)	13.83 (3.69)	11.71 (2.83)
Christian	13.84 (1.77)	11.81 (3.14)	11.62 (4.34)	10.29 (3.60)
Marital status				
Single	13.45 (2.70)	12.90 (2.77)	13.47 (3.44)	11.83 (2.64)
Married	13.34 (2.84)	12.86 (3.04)	14.15 (3.78)	11.57 (2.94)
Divorced	12.97 (2.40)	12.30 (3.36)	13.12 (4.26)	11.16 (3.23)
Widow	12.69 (2.17)	13.34 (2.51)	14.81 (2.87)	11.66 (2.47)
Has children				
Yes	13.43 (2.81)	13.01 (2.99)	14.17 (3.68)	11.64 (3)
No	13.43 (2.75)	12.91 (2.92)	13.54 (3.70)	11.81 (2.70)
Member of political party that aligns with personal values				
Yes	13.12 (2.80)	12.33 (3.12)	13 (4.13)	11.16 (2.95)
No	13.49 (2.69)	13.10 (2.79)	14.06 (3.45)	11.85 (2.75)
Political party membership				
Fateh	13.38 (2.67)	12.52 (3.05)	13.14 (4.14)	11.24 (2.77)
Hamas/Jehad	13.16 (2.96)	11.49 (3.59)	12.53 (4.48)	11.51 (3.59)
Left parties (Democratic Front, Popular Front, and Palestinian Front)	12.35 (3.00)	12.80 (2.85)	13.07 (4.36)	11.97 (2.50)
Others, independent, and cannot tell	11.71 (2.31)	13.17 (3.05)	13 (4.41)	9.38 (3.47)
Area of residence				
North	13.1 (2.81)	12.6 (2.80)	13.5 (3.70)	11.8 (2.71)
Middle Area	13.7 (2.42)	13.6 (2.93)	14.2 (3.63)	11.6 (2.59)
South	13.6 (2.84)	12.9 (3.03)	14.0 (3.65)	11.7 (3.12)
Highest level of education completed				
Primary school	11.80 (2.51)	10.85 (2.02)	10.91 (3.36)	10.84 (3.07)
Middle school	13.39 (2.85)	12.92 (2.82)	13.13 (4.23)	12.23 (2.50)
High school	13.51 (2.59)	12.97 (2.71)	13.81 (3.80)	11.52 (2.68)
Diploma or college	13.18 (2.75)	12.62 (2.72)	13.85 (3.50)	11.51 (3.04)
University bachelor’s degree	13.42 (2.79)	12.91 (3.03)	13.92 (3.70)	11.82 (2.77)
Technical or community college	14.19 (2.35)	13.78 (1.96)	14.44 (2.85)	11.19 (2.75)
Masters	13.02 (2.66)	13.43 (3.45)	14.47 (2.68)	11.93 (3.25)
PhD	20	19.33	20	20
Employment status				
Unemployed	13.45 (2.76)	12.93 (2.85)	13.96 (3.61)	11.54 (2.81)
Retired	13.15 (2.18)	12.06 (1.90)	12 (3.44)	10.59 (2.23)
Full-time government job	14.27 (2.39)	13.99 (2.87)	14.81 (3.37)	12.58 (2.52)
Part-time government job	12.82 (2.31)	13.02 (2.35)	13.83 (3.37)	12.44 (3.10)
Full-time job (private sector)	13.40 (2.78)	12.57 (3.10)	14.13 (3.55)	12.08 (2.48)
Special and part-time job (private or personal)	12.87 (2.21)	12.23 (3.28)	12.10 (3.56)	11.09 (2.98)
Perception of living in a peaceful area				
Yes	13.70 (2.75)	12.93 (3.02)	13.56 (4.00)	12.19 (2.87)
No	13.19 (2.74)	12.92 (2.87)	14.15 (3.38)	11.31 (2.79)

**Table 3 ijerph-20-06599-t003:** Association of perceived ingroup threat and perceptions of quality of life in the four domains measured using the WHOQOL-Bref.

WHO Domains	Physical Health Domain (n = 632)		Psychological Health Domain (n = 630)		Social Relationships Domain (n = 629)		**Environment Domain** **(n = 632)**	
	Coefficient	(95% CI)	Coefficient	(95% CI)	Coefficient	(95% CI)	Coefficient	(95% CI)
Demographic characteristics								
Ingroup	−0.060 **	−0.082, −0.039	−0.069 **	−0.091, −0.046	−0.062 **	−0.090, −0.034	−0.041 **	−0.064, −0.019
Age	−0.024	−0.055, 0.006	0.006	−0.026, 0.037	−0.033	−0.072, 0.006	−0.021	−0.053, 0.010
Sex								
Male	ref		ref		ref		ref	
Female	0.421	−0.056, 0.898	0.510 *	0.011, 1.009	0.878 *	0.264, 1.493	0.189	−0.306, 0.685
Religion								
Muslim	ref		ref		ref		ref	
Other	0.796	−1.587, 3.178	−0.916	−3.207, 1.375	−2.620	−5.448, 0.208	−1.434	−3.714, 0.846
Marital status								
Single	ref		ref		ref		ref	
Married	−0.201	−0.812, 0.410	−0.297	−0.935, 0.341	0.712	−0.076, 1.499	−0.307	−0.94, 0.33
Divorced	−0.657	−1.872, 0.558	−0.156	−1.414, 1.102	0.282	−1.271, 1.835	0.183	−1.07, 1.43
Widow	−0.714	−2.088, 0.660	0.156	−1.277, 1.589	0.766	−1.003, 2.534	0.221	−1.21, 1.65
Has children								
Yes	ref		ref		ref		ref	
No	0.034	−0.552, 0.620	−0.020	−0.063, 0.591	−0.385	−1.140, 0.370	−0.060	−0.67, 0.55
Member of political party that aligns with personal values								
Yes	ref		ref		ref		ref	
No	−0.261	−0.814, 0.291	0.336	−0.242, 0.913	0.027	−0.680, 0.735	0.463	−0.11, 1.03
Political party membership								
Fateh	ref		ref		ref		ref	
Hamas/Jehad	−0.691	−1.796, 0.413	−1.291	−2.443, −0.139	−1.314	−2.734, 0.106	−0.091	−1.24, 1.05
Left parties (Democratic Front, Popular Front, and Palestinian Front)	−0.942	−2.380, 0.495	−0.357	−1.858, 1.144	−1.182	−3.032, 0.669	0.495	−0.99, 1.98
Others, independent, and cannot tell	−0.686	−3.455, 2.083	1.948	−0.941, 4.84	0.394	−3.171, 3.959	−0.884	−3.76, 1.99
Area of residence								
North	ref		ref		ref		ref	
Middle Area	0.405	−0.223, 1.033	0.788 *	0.131, 1.44	0.722	−0.087, 1.532	−0.145	−0.796, 0.506
South	0.590 *	0.099, 1.081	0.514 *	0.001, 1.03	0.382	−0.250, 1.014	0.026	−0.483, 0.536
Highest level of education completed								
Primary school	ref		ref		ref		ref	
Middle school	0.429	−1.020, 1.878	0.514	−0.952, 1.98	0.538	−1.300, 2.376	1.233	−0.248, 2.713
High school	−0.021	−0.931, 0.890	−0.146	−1.102, 0.810	−0.059	−1.230, 1.112	−0.079	−1.02, 0.865
Diploma or college	−0.148	−1.103, 0.808	−0.158	−1.161, 0.845	0.370	−0.867, 1.607	0.054	−0.937, 1.05
University bachelor’s degree	−0.067	−0.901, 0.768	−0.134	−1.010, 0.743	0.103	−0.97, 1.18	0.078	−0.787, 0.944
Technical or community college	0.844	−1.664, 3.353	0.724	−1.895, 3.343	0.271	−2.958, 3.500	−0.353	−2.96, 2.25
Masters	−0.499	−1.998, 1.000	0.233	−1.334, 1.801	0.445	−1.484, 2.373	0.070	−1.48, 1.63
PhD	4.515	−0.924, 9.953	4.151	−1.523, 9.824	4.533	−2.468, 11.535	6.663 *	1.018, 1.625
Employment status								
Unemployed	ref		ref		ref		ref	
Retired	1.065	−0.890, 3.021	−1.210	−3.24, 0.817	−0.575	−3.078, 1.28	−1.473	−3.491, 0.545
Full-time government job	1.186 *	0.334, 2.038	1.256	0.369, 2.14	1.033	−0.062, 2.128	0.959 *	0.076, 1.841
Part-time government job	−0.207	−1.740, 1.325	0.650	−1.010, 2.309	0.419	−1.553, 2.392	1.249	−0.341, 2.839
Full-time job (private sector)	0.224	−0.619, 1.066	−0.205	−1.081, 0.671	0.393	−0.688, 1.473	0.514	−0.357, 1.3986
Special and part-time job (private or personal)	−0.443	−1.303, 0.417	−0.292	−1.189, 0.606	−1.635 *	−2.740, −0.530	−0.416	−1.308, 0.476
Perception of living in a peaceful area								
Yes	ref		ref		ref		ref	
No	−0.355	−0.800, 0.090	−0.013	−0.477, 0.451	0.709 *	0.134, 1.284	−0.704 *	−1.166, −0.242
Adjusted R square	0.064		0.085		0.085		0.050	
F statistic	2.59		3.18		3.17		2.23	
*p*-value	<0.001		<0.001		<0.001		<0.001	

Note: *B* = Unstandardized regression coefficient. * Indicates significance at *p* < 0.05, ** indicates significance at *p* < 0.001.

**Table 4 ijerph-20-06599-t004:** Association of perceived outgroup threat and demographic characteristics in relation to the four WHO Domains.

WHO Domains	Physical Health Domain (n = 640)		Psychological Health Domain (n = 638)		Social Relationships Domain (n = 638)		Environment Domain (n = 639)	
	Coefficient	(95% CI)	Coefficient	(95% CI)	Coefficient	(95% CI)	Coefficient	(95% CI)
Demographic characteristics								
Outgroup	0.050 **	0.028, 0.072	0.046 **	0.023, 0.070	0.069 **	0.040, 0.098	0.005	−0.019, 0.028
Age	−0.017	−0.046, 0.013	0.005	−0.026, 0.036	−0.033	−0.071, 0.005	−0.024	−0.055, 0.007
Sex								
Male	ref		ref		ref		ref	
Female	0.265	−0.207, 0.737	0.411	−0.093, 0.916	0.664 *	0.048, 1.28	0.101	−0.395, 0.597
Religion								
Muslim	ref		ref		ref		ref	
Other	0.091	−2.50, 2.681	−1.421	−3.938, 1.097	−3.454 *	−6.533, −0.374	−1.034	−3.514, 1.446
Marital status								
Single	ref		ref		ref		ref	
Married	−0.072	−0.667, 0.522	−0.199	−0.834, 0.437	0.746 *	−0.031, 1.523	−0.262	0.888, 0.364
Divorced	−0.478	−1.655, 0.699	−0.321	−1.566, 0.925	0.108	−1.415, 1.632	−0.193	−1.420, 1.034
Widow	−0.824	−2.134, 0.486	0.012	−1.387, 1.411	1.039	−0.671, 2.749	0.258	−1.12, 1.64
Has children								
Yes	ref		ref		ref		ref	
No	0.064	−0.508, 0.636	−0.046	−0.657, 0.565	−0.491	−1.238, 0.256	−0.072	−0.673, 0.529
Member of political party that aligns with personal values								
Yes	ref		ref		ref		ref	
No	−0.017	−0.561, 0.528	0.427	−0.156, 1.009	0.181	−0.527, 0.888	0.385	−0.185, 0.955
Political party membership								
Fateh	ref		ref		ref		ref	
Hamas/Jehad	−0.402	−1.482, 0.677	−1.145	−2.298, 0.008	−0.926	−2.334, 0.483	−0.315	−1.450, 0.820
Left parties (Democratic Front, Popular Front, and Palestinian Front)	−1.105	−2.487, 0.277	0.051	−1.425, 1.528	−0.652	−2.456, 1.152	0.667	−0.786, 2.120
Others, independent, and cannot tell	−1.143	−3.873, 1.598	1.265	−1.652, 4.182	−0.012	−3.578, 3.555	−1.623	−4.496, 1.250
Area of residence								
North	ref		ref		ref		ref	
Middle Area	0.521	−0.089, 1.132	0.979 *	0.326, 1.632	0.857	0.059, 1.655	−0.098	−0.742, 0.545
South	0.290	−0.203, 0.782	0.213	−0.314, 0.741	0.106	−0.537, 0.749	−0.026	−0.544, 0.492
Highest level of education completed								
Primary school	ref		ref		ref		ref	
Middle school	0.126	−1.218, 1.470	0.329	−1.074, 1.732	0.255	−1.480, 1.991	0.841	−0.555, 2.238
High school	−0.055	−0.965, 0.854	−0.237	−1.215, 0.740	0.048	−1.139, 1.235	−0.266	−1.223, 0.691
Diploma or college	−0.143	−1.108, 0.821	−0.160	−1.196, 0.877	0.173	−1.091, 1.438	−0.085	−1.099, 0.930
University bachelor’s degree	−0.060	−0.895, 0.775	−0.245	−1.144, 0.653	0.167	−0.923, 1.257	0.018	−0.860, 0.896
Technical or community college	0.834	−1.473, 3.141	0.730	−1.736, 3.197	0.864	−2.150, 3.877	−0.343	−2.770, 2.085
Masters	−0.255	−1.744, 1.234	0.356	−1.238, 1.950	0.654	−1.29, 2.59	0.075	−1.491, 1.640
PhD	4.634	−0.770, 10.038	4.387	−1.386, 10.159	4.692	−2.367, 11.75	6.808 *	1.122, 12.495
Employment status								
Unemployed	ref		ref		ref		ref	
Retired	1.048	−0.796, 2.893	−0.967	−2.925, 0.990	−0.606	−3.002, 1.790	−0.788	−2.719, 1.144
Full-time government job	1.242 *	0.394, 2.091	1.478 *	0.573, 2.384	1.033	−0.074, 2.139	1.260 *	0.368, 2.152
Part-time government job	−0.466	−1.933, 1.00	0.443	−1.176, 2.062	0.062	−1.851, 1.976	0.870	−0.672, 2.411
Full-time job (private sector)	0.369	−0.483, 1.220	−0.013	−0.921, 0.894	0.758	−0.351, 1.867	0.590	−0.303, 1.484
Special and part- time job (private or personal)	−0.438	−1.295, 0.419	−0.241	−1.156, 0.675	−1.517 *	−2.636, −0.399	−0.557	−1.459, 0.344
Perception of living in a peaceful area								
Yes	ref		ref		ref		ref	
No	−0.644	−1.090, −0.197	−0.185	−0.662, 0.292	0.448	−0.136, 1.03	−0.747 *	−1.217, −0.276
Adjusted R square	0.047		0.054		0.086		0.034	
F statistic	2.16		2.34		3.22		1.83	
p-value	<0.001		<0.001		<0.001		<0.001	

Note: B = Unstandardized regression coefficient; * Indicates significance at *p* < 0.05, ** indicates significance at *p* < 0.001.

## Data Availability

The data are not publicly available due to ongoing research.

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
