# Peer review of "Exploring the Influence of Perceived Ingroup and Outgroup Threat on Quality of Life in a Region Impacted by Protracted Conflict"

_ijerph, 2023, doi:10.3390/ijerph20166599_

Round 1
Reviewer 1 Report
Thanks for performing this interesting study dealing with the very sensitive topic of quality of life in populations affected by protracted conflicts. I have some suggestions that could improve your work:
1) abstract: clarify the distinction between "ingroup threat" and "outgroup threat"; keywords: "ingroup threat" and "outgroup threat" should be closed
2) the introduction, despite well-written and rich of valuable information, could be re-organized to make clearer the distinction between "ingroup threat" and "outgroup threat" that are key concepts of this study
3) lines 67-70: please rephrase these four lines as they are not fully understandable (at least for non-expert readers). For example, what "biases" are you referring to? What do you mean for "similarities" and "favoritisms"?
4) lines 99-64: please rephrase since this sentence is too long/complex
5) par 2.1: provide more details on sample recruitement and survey administration process (type/structure of survey, recruitment process, mode of administration, response rate, etc.)
6) par. 2.5: provide more details on the multivariable regression performed (e.g., type of model, goodness-of-fit measures, software used, etc.)
7) lines 194-195: data on educational level should be separated from data on employment status.
8) Moreover, in the discussion, you could provide some interpretation of this phenomenon in relation to the conflict or, more generally, the socio-economic situation of the area (i.e., despite almost half of participants have a university degree, the majority is unemployed)
9) par. 3.1-3.4: these paragraphs should be collapsed into a single paragraph
10) par. 3.5-3.6: the title is too long (you could delate the second part after "quality of life")
11) Tables 2, 3, 4: you should provide some comments in the text also in relation to the impact of socio-demographic characteristics on quality of life
12) Tables 3-4: the title should be revised since these tables should provide the results of multivariate regression models, according to the methods. Also, please refer to the literature to adopt the standard way of reporting regression models results (for example, spaces in the table are excessive, it would be better to report each model on a single page)
13) lines 348-351: the second part of the sentence is not fully understandable (please clarify how outgroup threat can act as a double-edged sword)
14) lines 368-370: please rephrase this sentence (what do you mean for "negative contact")?
15) lines 401-403: the last sentence is too vague (e.g., what differences are you referring to)?
None
Reviewer 2 Report
Thank you for providing an opportunity to review the manuscript. This is an interesting study that explored the Influence of Perceived Ingroup and Outgroup Threat on Quality of Life in a Region Impacted by Protracted Conflict. The text is relatively well written; however, it needs major improvement:
1) In the abstract, please provide the study design, the number of participants, and analysis methods, such as regression.
2) In the introduction, please add more references from previous studies on factors related to quality of life, which will be used in the regression analysis. Additionally, include more information about perceptions of threat from the ingroup and perceptions of quality of life in the four domains from previous or related studies.
3) In the introduction, please mention previous studies on the WHOQOL-Bref.
4) In the materials and methods, please provide more details about the inclusion and exclusion criteria.
5) In the materials and methods, please explain how to calculate the sample size and how to randomly sample participants. Additionally, include the relevant reference for calculating the sample size.
6) In the analysis, please explain the assumptions of the regression analysis and the process for selecting each variable in the multivariable linear regression models. Also, please specify the alpha value for hypothesis testing.
7) Extensive editing of English language and style is required.
8) In the results, some confidence interval (CI) results are missing from the table.
9) In the results, please interpret and present all the factors associated with the WHOQOL-BREF domains, such as area of residence and employment status
10) In the discussion, please discuss the descriptive results with previous studies, such as the mean score of quality of life (QOL) between this study and previous studies. Additionally, include a discussion about factors other than ingroup and outgroup that are related to QOL.
Extensive editing of English language and style is required.
Round 2
Reviewer 1 Report
I thank the authors for addressing carefully my previous comments. I still have few minor comments:
- I don't think it is worth to split the introduction into three separate paragraphs.
- In tables 3-4, I don't understand why the sample size is reported only for physical health domain.
- The sample size (n=709) and actual response rate should be reported in par. 5 (results)
Author Response
Dear Reviewer, thank you for your final comments. We appreciate your time and feedback on this important work, and for considering our manuscript for publication. Please see our responses to your final comments below:
- I don't think it is worth splitting the introduction into three separate paragraphs.
Response: We agree and have removed the three subsection divisions for a better flow of the introduction.
- In tables 3-4, I don't understand why the sample size is reported only for the physical health domain.
Response: We apologize for leaving out this detail. We have changed this to include the sample size for each domain model in Tables 3-4.
- The sample size (n=709) and actual response rate should be reported in par. 5 (results)
Response: This detail has been moved to the results section as suggested. Please review Lines: 262-263.
Reviewer 2 Report
Thank you for giving me the chance to review the manuscript. This study, titled "Influence of Perceived Ingroup and Outgroup Threat on Quality of Life in a Region Impacted by Protracted Conflict," is highly engaging and sheds light on a significant topic. The overall text is well written, and I commend the authors for their remarkable efforts in enhancing the manuscript. All the suggested revisions have been addressed, and I believe it is now ready for publication.
Author Response
We thank the reviewer for their time in reviewing this important manuscript, and their consideration for recommending our manuscript for publication. We are very pleased that the enhanced manuscript is now ready for publication.